# OpenReview forum: "SMART-3D: Scaling Masked AutoRegressive Transformer for Efficient 3D Shape Generation"
_ICLR.cc/2026/Conference — ICLR 2026 Conference Withdrawn Submission_

### Official Review · Reviewer_zn3H · 2025-10-30

**Soundness:** 1
**Presentation:** 2
**Contribution:** 1
**Rating:** 2
**Confidence:** 4

**Summary:**

The paper proposes SMART-3D, a mask token modeling approach for 3D generation.

**Strengths:**

The paper studies a fundamental problem in the 3D domain, 3D generation.

**Weaknesses:**

There are obvious flaws in the interpretation of metrics and methods:

1. The dimension in the equation (line 244) is not right. Here, $\mathbf{B}_{\text{spatial}} \in \mathbb{R}^{L \times L}$ while $\phi\left(\mathbf{Q}\right)\left(\phi\left(\mathbf{K}\right)^{T}\mathbf{V}\right)  \in \mathbb{R}^{L \times d}$. Therefore, they can not be added.

2. The metrics in Table 1 is not right. The optimum score for `1-NNA` should be 50\%, *i.e.* the nearest neighbor classifier cannot distinguish the generated samples from the ground truth samples, not "the lower the better".

3. There are no ablation studies on the highlighted architectures, *e.g.* the impact of linear attention, the generation speed comparison with other diffusion / auto-regressive methods.

**Questions:**

See weakness part.

---

### Official Review · Reviewer_kuxL · 2025-11-01

**Soundness:** 2
**Presentation:** 3
**Contribution:** 1
**Rating:** 2
**Confidence:** 5

**Summary:**

The paper proposes an framework that merges masked autoregressive generation with diffusion modeling and linear attention, addressing key efficiency bottlenecks in 3D shape generation. However, technically novelty and evaluation are limited.

**Strengths:**

The progressive masked decoding strategy combined with linear attention (O(L) complexity vs O(L²)) enables faster inference (up to 3.2× speedup) while maintaining quality.

The paper includes thorough experiments covering unconditional generation, conditional completion, multi-class scaling, and model size ablations, demonstrating robustness across different settings.

**Weaknesses:**

Limited novelty in individual components: While the combination is novel, the core techniques (masked autoregressive models, linear attention, diffusion processes) are well-established. The paper primarily integrates existing ideas rather than introducing fundamentally new concepts.

The paper doesn't compare against some recent 3D generation methods, particularly newer diffusion-based approaches and more recent autoregressive models shown in the Missing refrences below. The baseline comparisons are somewhat dated (mostly 2021-2023).

Missing refrences:
TRELLIS: Structured 3D Latents for Scalable and Versatile 3D Generation. In CVPR 2025
SAR3D: Autoregressive 3D Object Generation and Understanding via Multi-scale 3D VQVAE. In CVPR 2025
MAR-3D: Progressive Masked Auto-regressor for High-Resolution 3D Generation. In CVPR 2025
G3PT: Unleash the Power of Autoregressive Modeling in 3D Generation via
Cross-Scale Querying Transformer. In IJCAI 2025
Cube3D:https://arxiv.org/abs/2503.15475

**Questions:**

You mention hierarchical token representation, but how is this implemented? Are there multiple resolution levels? How do they interact?

Can you provide detailed time and memory consumption comparisons with baselines？

How does the model perform on out-of-distribution shapes or novel object categories not seen during training?

How this formulation is different from MAR-3D: Progressive Masked Auto-regressor for High-Resolution 3D Generation. In CVPR 2025?

---

### Official Review · Reviewer_jvkj · 2025-11-03

**Soundness:** 3
**Presentation:** 3
**Contribution:** 2
**Rating:** 4
**Confidence:** 4

**Summary:**

This paper introduces SMART-3D (Scaling Masked AutoRegressive Transformers for 3D generation) for 3D shape generation. The framework combines the modeling capability of autoregressive models with the efficiency of masked generation strategies. It uses progressive masked decoding to enable parallel decoding and reduce sampling steps, and employs a linear attention mechanism to lower computational complexity, achieving state-of-the-art performance in both generation quality and speed.

**Strengths:**

+ The motivation of this paper is solid, and the technique sounds interesting
+ Designed a progressive masking strategy, predicting multiple tokens in parallel to improve generation speed.
+ By using a linear attention mechanism, memory usage is reduced, and positional information is incorporated into the computation through 3D biases.

**Weaknesses:**

- The paper lacks an illustration of a complete model framework and does not show how category labels are fed into the network, which affects the clarity of the method.
- The progressive masking strategy of the paper provides multiple approaches (diagonal, spiral), but does not compare the effects brought by different masking strategies.
- Table 1 is incorrectly labeled. The FastDiT-3D-S method performs best in the fourth column for the object 'chair', which contradicts the table title stating that it 'significantly outperforms previous baselines in all categories.
- It lacks comparisons with other methods published in 2024-2025 (such as DiffGS [A]). The evaluation of the method's effectiveness is insufficient.

[A] Zhou, J., Zhang, W., & Liu, Y.-S. (2024). DiffGS: Functional Gaussian Splatting Diffusion

**Questions:**

- What masking strategy was used in the final experiment? Do different masking strategies affect the model's performance?
- Where exactly is the learnable class embedding applied in the model? Is it implicitly used as a condition in the cross-attention mechanism in SMART-3D?
- Why weren't comparisons with the latest methods included, and all the comparison methods are from 2023 or earlier?

---

### Official Review · Reviewer_nxya · 2025-11-03

**Soundness:** 1
**Presentation:** 1
**Contribution:** 1
**Rating:** 0
**Confidence:** 5

**Summary:**

-

**Strengths:**

-

**Weaknesses:**

This paper does not show any visual comparison with baselines.

The author seems to misunderstand the concept of an autoregressive model.

This is an outdated paper that only compares with methods from before 2023.

**Questions:**

-

---

### Note · Authors · 2025-11-27

I have read and agree with the venue's withdrawal policy on behalf of myself and my co-authors.